# TMAO and Cardiovascular Disease: Exploring Its Potential as a Biomarker

**DOI:** 10.3390/medicina61101767

**Published:** 2025-09-30

**Authors:** Octavian Amaritei, Oana Laura Mierlan, Ciprian Adrian Dinu, Iulia Chiscop, Madalina Nicoleta Matei, Cristian Gutu, Gabriela Gurau

**Affiliations:** 1Faculty of Medicine and Pharmacy, University “Dunărea de Jos” Galați, 800008 Galați, Romania; octavian.amaritei@ugal.ro (O.A.); ciprian.dinu@ugal.ro (C.A.D.); iulia.chiscop@ugal.ro (I.C.); madalina.matei@ugal.ro (M.N.M.); gabriela.gurau@ugal.ro (G.G.); 2“Sf. Andrei” Clinical Emergency County Hospital, 800578 Galați, Romania; 3“Sf. Ioan” Emergency Clinical Pediatric Hospital, 800487 Galați, Romania; 4Center for Research and Technology Transfer in the Medico-Pharmaceutical Field, “Dunărea de Jos” University of Galați, 800008 Galați, Romania; 5“Dr. Aristide Serfioti” Emergency Military Hospital Galați, 800150 Galati, Romania

**Keywords:** gut microbiome, TMAO, choline, cardiovascular disease, atherosclerosis

## Abstract

Gut microbiota has increasingly been shown to exert effects beyond the gastrointestinal tract, some of which are mediated through its metabolites, such as trimethylamine N-oxide (TMAO)—a compound converted by gut bacteria from dietary choline found predominantly in animal products that is associated with cardiovascular disease (CVD). However, a significant gap persists in human clinical trials assessing its potential causal role. This narrative review aims to present the current understanding of the gut microbiome, TMAO, and their relationship with CVD, while proposing future directions that may support the use of TMAO as a biomarker and guide potential interventions to reduce its harmful impact. Both animal and human studies have demonstrated a link between TMAO and CVD, with animal studies also indicating a causal effect—showing increased cardiovascular risk following TMAO administration and reduced risk when TMAO is eliminated. While direct extrapolation from animal models to humans is limited due to biological differences, these findings offer a foundation for the development of well-designed clinical trials in human populations. Although direct approaches to target TMAO—such as trimethylamine (TMA) lyase inhibitors and antisense oligonucleotide (ASO) therapy—have shown promising results in animal studies, they have yet to be investigated in human trials, leaving indirect strategies such as dietary changes and probiotics as the only currently available options.

## 1. Introduction

Cardiovascular disease remains the leading cause of mortality worldwide [1]. Although considerable research has been dedicated to optimizing the management of established CVD, greater emphasis must be placed on preventive strategies, which hold significant potential for improving patient outcomes and reducing healthcare burden [2].

Recent discoveries have drawn attention to the gut microbiome as a critical, yet underexplored, contributor to cardiovascular health [3,4,5]. In particular, TMAO—a metabolite generated through microbial metabolism of dietary precursors such as choline and carnitine—has emerged as a potential mediator linking gut dysbiosis to cardiovascular pathology. Elevated TMAO levels have been associated with pro-inflammatory signaling, endothelial dysfunction, and enhanced atherogenesis, suggesting a contributing role in the pathophysiology of CVD [6,7].

Increases in TMAO are associated with negative alterations in gut microbiota, particularly in the ratio of beneficial to harmful bacteria, as well as with a higher incidence of CVD [8,9,10,11]. At the other end of the spectrum, studies in which TMAO levels decreased—either through dietary restriction or probiotic supplementation—have shown improvements in both the gut microbiome and cardiovascular outcomes [12,13]. While studies have attempted to isolate the effects of TMAO, we are still far from fully understanding its role in CVD [14].

The aim of this review is to explore the potential role of TMAO as a link between gut dysbiosis and cardiovascular disease, highlighting its promise as an accessible biomarker for both gut microbial alterations and cardiovascular risk.

## 2. Materials and Methods

### 2.1. Research Strategy

Keywords such as “gut microbiome”, “TMAO”, “cardiovascular disease”, “hypertension”, “metabolic diseases”, “atherosclerosis”, “heart failure”, and their combinations were used to perform a comprehensive literature search in PubMed, ScienceDirect, and Google Scholar. Additionally, information from journals (e.g., MDPI, Frontiers) and publishers (e.g., American Heart Journal) was included, as some articles were available only as abstracts in databases, with the full texts accessible exclusively through the journal or publisher websites. Both animal and human studies published within the last ten years were included to ensure the novelty of the research, with the timeframe extended in a few cases to incorporate studies offering valuable insights into underexplored factors. The first section of this review presents information on TMAO biochemistry and its involvement in the pathophysiological mechanisms underlying CVD, while the second section explores specific associations between TMAO levels and individual cardiovascular conditions.

Given the narrative nature of this review, no formal risk of bias tools were applied. References were organized using Zotero 7.0.26, and artificial intelligence tools were employed to check grammar and ensure clarity and coherence of the text.

### 2.2. Inclusion and Exclusion Criteria

Study selection was conducted according to predefined inclusion and exclusion criteria. Only peer-reviewed studies addressing TMAO, the gut microbiome, and CVD were considered. Exclusion criteria included non-peer-reviewed sources, duplicate publications, and those with incomplete data. Also, studies with major limitations and older than the specified period that did not provide additional information were excluded.

### 2.3. Study Selection

The initial search generated 3793 results. After screening titles and abstracts, 74 studies were selected for inclusion based on their relevance and novelty. However, certain key information regarding TMAO’s metabolism and biochemistry remained absent. To fill these gaps, the search timeframe was extended by six years (2009–2025), leading to the addition of six more studies, the earliest of which was published in 2009. After the initial review, 13 more studies were added, for a total of 93 studies.

## 3. Trimethylamine N-Oxide (TMAO)

### 3.1. Origin and Metabolism of TMAO

TMAO is a dietary-derived amine oxide, formed as an oxidized product of TMA. In the gut, choline, betaine, and L-carnitine are converted by the bacteria into TMA, which is absorbed into the bloodstream and directed mostly towards the liver, where it becomes oxidized through hepatic flavin monooxygenases (FMO) to TMAO [15]. Sources of dietary carnitine and choline include animal products, especially eggs, red meat, fish, and dairy (Figure 1) [16]. A small percentage of TMA is also converted locally, in the intestine, to ammonia, methylamine, and dimethylamine [8]. TMAO is mostly excreted through urine, with a very small percentage from sweat, stool, and respiration [17].

While at first glance TMAO may appear to be merely a gut-derived metabolite, this might suggest that its effects are limited to the digestive system. However, similar to other metabolites—such as short chain fatty acids (SCFAs)—TMAO has significant implications for various organ systems, including the heart, blood vessels, kidneys, and brain [18,19]. Regarding the cardiovascular system, evidence shows that TMAO influences hypertension, atherosclerosis, metabolic disorders, and heart failure [14,20,21].

### 3.2. Gut Microbiota, TMAO, and CVD

TMAO promotes endothelial dysfunction by reducing the activity of endothelial nitric oxide synthase (eNOS) and the production of nitric oxide (NO). Furthermore, it also activates protein kinase C (PKC), which further stimulates the nuclear factor kappa-light-chain-enhancer of activated B cells (NF-κB) pathway. These changes lead to the release of inflammatory cytokines, such as interleukin-1 (IL-1), IL-18, and tumor necrosis factor α (TNF-α), and upregulation of vascular cell adhesion molecule-1 (VCAM-1), which increases monocyte adhesion [22,23].

TMAO is also involved in three other processes that set the ground for the progression of vascular inflammation, atherosclerosis, and its complications. Firstly, TMAO inhibits reverse cholesterol transport (RCT), by which excess cholesterol is normally removed from peripheral tissues and returned to the liver for excretion, and upregulates proprotein convertase subtilisin/kexin type 9 (PCSK9), an enzyme that degrades low-density lipoprotein (LDL) cholesterol receptors. The end result of these two processes is a decrease in the liver uptake of cholesterol and, consequently, more circulating LDL cholesterol [24,25]. Secondly, TMAO promotes the expression of scavenger receptors such as cluster of differentiation 36 (CD36) on macrophages, which increases their cholesterol uptake and leads to the development of foam cells [26,27]. Last but not least, TMAO increases adenosine diphosphate (ADP)–induced platelet reactivity [28] (Figure 2).

### 3.3. Mendelian Randomization Studies

Mendelian randomization (MR) studies have been conducted to explore the causal relationship between TMAO and CVD. A 2022 MR study using genetic data from 2076 European participants suggested that genetically predicted higher levels of TMAO and carnitine were associated with increased systolic blood pressure. Yet, this association lost statistical significance in MR-Egger sensitivity analyses, possibly due to horizontal pleiotropy. The limited ethnic diversity of the sample further restricts the applicability of the results [29].

Jia et al. conducted an MR study that did not find a significant causal relationship between genetically predicted higher TMAO levels and various cardiometabolic diseases, including type 2 diabetes, atrial fibrillation, coronary artery disease, myocardial infarction, stroke, and chronic kidney disease [30].

While human and animal studies provide evidence of associations between TMAO and cardiovascular disease, MR studies—currently scarce, with only a few available—generally do not support a causal relationship. These discrepancies may reflect a truly non-causal association, differences in population, study design, sample sizes, and methodologies, or limitations inherent to MR studies, such as reliance on valid genetic instruments, potential pleiotropy, insufficient statistical power, and population stratification, which can obscure causal inference. Further well-powered MR studies are needed to clarify this relationship.

## 4. Hypertension

### 4.1. Dietary Patterns and Blood Pressure

Hypertension is the primary contributor to cardiovascular disease and early mortality on a global scale [31]. The relationship between blood pressure, individual foods, and dietary patterns—such as the Mediterranean or the Dietary Approaches to Stop Hypertension (DASH) diet—has been explored in many studies, with most showing similar results: diet is one of the most impactful elements in blood pressure regulation [32,33,34,35]. Emerging evidence indicates that, beyond salt intake, dietary components such as polyphenols, fiber, and omega-3 fatty acids significantly influence blood pressure regulation [36,37]. Moreover, these factors appear to interact with TMAO levels, as low fiber and high salt intake are associated with increased TMAO concentrations [38,39].

### 4.2. Animal Models: Angiotensin-II, Aging and Vascular Stiffness

Owing to their more invasive nature, animal models can offer valuable mechanistic insights and lay the foundation for further research. Jiang et al. [40] showed through plasma samples, imagistic and histopathologic tests that, in mice, TMAO worsens angiotensin II (AT-II)–induced hypertension by aggravating mesenteric and afferent arteriole vasoconstriction, which also led to a decrease in glomerular filtration rate. These changes were reversed after the administration of antibiotics. Brunt et al. [41] showed that in both humans and mice, TMAO levels increase with age and are positively associated with rising blood pressure. It is well known that aging contributes to vascular stiffening and hypertension, so these findings stop short of establishing causality [42]. Importantly, the study extended its analysis by supplementing mice with dietary TMAO and monitoring aortic stiffness. TMAO supplementation increased aortic stiffness in young mice and exacerbated it in older ones, effects that were partially mediated by the accumulation of advanced glycation end products (AGEs) and superoxide-induced oxidative stress. These findings suggest that dietary TMAO may contribute to hypertension, at least in part through its effects on vascular stiffening.

### 4.3. Human Studies and Genetic Evidence

A meta-analysis of eight studies with a total of 11750 subjects from multiple geographical areas, published in 2020, suggested a possible association between TMAO and hypertension but could not clearly elucidate the underlying mechanisms involved, instead postulating several pathways involving inflammation, atherosclerosis, and the microbiome. While the authors selected high-quality studies, information regarding dietary intake was lacking. Furthermore, all included studies were conducted in populations with high cardiovascular risk, which differs from healthy individuals—limiting the generalizability of the findings to the broader population, who would arguably benefit the most from preventive strategies [43]. Future prospective studies including individuals with lower cardiovascular risk, accounting for dietary intake, standardized TMAO sampling, and renal function adjustments, are needed.

## 5. Atherosclerosis

### 5.1. Atherosclerosis, TMAO and Inflammation

Atherosclerosis is a systemic, progressive condition responsible for the majority of cardiovascular-related deaths worldwide [44]. While dyslipidemia remains a key risk factor, emerging evidence underscores the role of inflammation in its pathogenesis [45]. The gut microbiota, together with its metabolites, may play a significant role in modulating the course of atherosclerosis through their influence on cholesterol metabolism and inflammatory pathways [46].

Wang et al. [13] demonstrated that, in a mouse model of atherosclerosis, dietary supplementation with choline or TMAO significantly increased aortic plaque size without affecting plasma lipids, glucose, or hepatic fat. Elevated TMAO levels, rather than choline, correlated with plaque burden in both male and female mice, with females showing higher baseline TMAO levels. These findings suggest a potential causal role for TMAO in atherosclerosis, independent of traditional lipid parameters.

In a nested case–control study, plasma TMAO and related metabolites, quantified by stable isotope dilution LC-MS/MS, demonstrated a dose-dependent association with plaque burden, independent of gender [13]. Associations with CAD, peripheral arterial disease, and ASCVD were evaluated after adjusting for key clinical variables, including age, sex, renal function (eGFR), fasting glucose, lipid profile, CRP, and troponin I.

In patients newly diagnosed with coronary artery disease (CAD)—most often resulting from atherosclerosis—TMAO levels were found to correlate with the SYNTAX score, a widely used marker of plaque burden and disease severity. This finding comes from a prospective observational single-center study including 429 subjects undergoing coronary angiography, with detailed collection of demographics, clinical history, and disease status, while excluding prior revascularization. Plasma TMAO was measured by triple quadrupole mass spectrometry, and associations with SYNTAX score were evaluated using linear and logistic regression, adjusting for age, sex, hypertension, diabetes, and hyperlipidemia [47]. While these findings are intriguing, TMAO cannot replace more invasive investigations such as coronary angiography in clinical practice, as its levels are influenced by multiple factors, including diet, renal function, and even diurnal variation [48,49].

### 5.2. TMAO and Plaque Instability

While there is evidence linking TMAO to the development of atherosclerotic cardiovascular disease (ASCVD), it is equally important to explore its potential role in disease progression, particularly given that atherosclerosis is a continuous process, with its clinical manifestations closely tied to plaque stability [50,51]. You and Gao [52] conducted a study involving 90 patients with CAD and 90 healthy controls to investigate the relationship between the gut microbiome, circulating biomarkers, and plaque stability. Blood samples were collected to assess levels of TMAO, phenylacetylglutamine (PAGln), and standard lipid parameters. Coronary plaques were evaluated using coronary angiography and intravascular ultrasound, while fecal samples were analyzed to characterize gut microbiota composition. As anticipated, patients with CHD exhibited elevated levels of total cholesterol, LDL-cholesterol, and triglycerides, along with significantly higher concentrations of TMAO and PAGln. Notably, a positive association was observed between TMAO levels and plaque vulnerability. Furthermore, TMAO was associated with plaque rupture in a prospective study by Tan et al. [53], which included 211 subjects presenting with ST-segment elevation myocardial infarction (STEMI) who underwent optical coherence tomography (OCT) imaging of culprit lesions, along with a second independent cohort of 53 healthy individuals providing reference TMAO values. Plasma TMAO was quantified by stable isotope dilution high-performance LC-MS/MS, and associations with plaque rupture were analyzed using Receiver Operating Characteristic (ROC) curves, adjusting for age, sex, smoking, triglycerides, diabetes, hypertension, high-sensitivity C-reactive protein (hs-CRP), and estimated glomerular filtration rate (eGFR). The authors also suggested that TMAO could serve as a biomarker for plaque rupture in such patients; however, the urgency of invasive management represents a significant limitation to its practical application [53].

### 5.3. TMAO and Major Adverse Cardiovascular Events (MACE)

Tang et al. [12] studied the relationship between TMAO levels and the risk of MACE during 3 years of follow-up in 4007 patients who underwent elective cardiac catheterization. Associations with outcomes were analyzed using Cox proportional-hazards regression and Kaplan–Meier analysis, adjusting for age, sex, systolic blood pressure (SBP), diabetes, lipid profile, triglycerides, smoking, hs-CRP, eGFR, leukocyte count, body mass index (BMI), medication history, and angiographic extent of coronary artery disease. The study found that higher fasting plasma levels of TMAO were associated with an increased risk of MACE. Participants in the highest TMAO quartile had a significantly greater event risk compared to those in the lowest quartile, with a clear, graded relationship observed between rising TMAO levels and cardiovascular risk.

## 6. Metabolic Diseases

### 6.1. Diet, Gut Microbiota, and Obesity

Metabolic diseases such as obesity are major cardiovascular risk factors [54]. These conditions are among the most likely to be influenced by food intake, as they are typically associated with a caloric surplus. However, the differences in gut microbial composition between normal-weight and obese individuals suggest that additional mechanisms may be involved [55,56,57,58].

### 6.2. TMAO and Obesity: Clinical Evidence

Evidence from cross-sectional observational studies suggests a potential link between TMAO and obesity. Barrea et al. [59] reported a positive association between circulating TMAO levels and increased body weight, insulin resistance, and hepatic steatosis in 137 adult subjects. Similarly, Mihuta et al. [60] observed comparable results in obese children, indicating a potential role for TMAO as an early biomarker of metabolic disturbances. However, the lack of data on gut microbiota composition, detailed nutrient intake, inflammatory markers, and TMAO precursors limits the ability to fully explore the underlying mechanisms involved.

Pescari et al. [61] conducted a study on 60 subjects to investigate the relationship between TMAO, obesity, and cardiovascular risk. Body composition was assessed through BMI and bioimpedance parameters, evidence of subclinical atherosclerosis was evaluated using carotid ultrasound, and blood markers indicative of metabolic disease—such as fasting blood glucose, hemoglobin A1C, and lipid profile—were analyzed. Although limited by the small sample size and geographic restriction, TMAO was correlated with increased intima-to-media thickness (IMT) in subjects with obesity, a marker of subclinical atherosclerosis. Additionally, the study found an association between TMAO levels and a family history of metabolic diseases, suggesting possible genetic mechanisms influencing its concentration. Higher TMAO levels were also observed in individuals more prone to sedentary behaviors. While the findings are promising, the study does not conclusively determine whether TMAO contributes causally to increased cardiovascular risk or simply reflects a state of low-grade chronic inflammation, which is representative of obesity.

## 7. Heart Failure

### 7.1. Heart Failure and TMAO: A Vicious Cycle

Heart failure (HF) is a complex clinical syndrome associated with high morbidity and mortality, reduced functional capacity and quality of life, and substantial healthcare costs [62,63]. Systemic congestion—a hallmark of HF—leads to increased intestinal permeability, which facilitates bacterial translocation and enhanced TMAO reabsorption, creating a vicious cycle that allows elevated TMAO levels to exert further detrimental effects [64].

Trøseid et al. [65] found higher circulating levels of choline, betaine, and TMAO in individuals with chronic heart failure. Although all three metabolites showed associations with clinical, hemodynamic, and neurohormonal markers of disease severity, only TMAO was independently linked to adverse outcomes during follow-up. Notably, TMAO levels were highest in patients with heart failure of ischemic origin, highlighting a potential connection with atherosclerosis.

### 7.2. TMAO: A Prognostic Factor in Heart Failure with Preserved Ejection Fraction (HFpEF)

Kinugasa et al. [66] investigated TMAO concentrations in a cohort of 146 patients hospitalized for acute decompensated heart failure with preserved ejection fraction (HFpEF). They observed that individuals with elevated TMAO levels had a higher frequency of prior hospitalizations. Over a mean follow-up period exceeding two years, mortality was significantly greater in the high-TMAO group (46.6%) compared to the low-TMAO group (27.4%). Elevated TMAO levels were independently associated with a higher incidence of the composite primary outcomes. However, the study also identified a significant interaction between TMAO and nutritional status, as measured by the Geriatric Nutritional Risk Index (GNRI): the adverse prognostic impact of elevated TMAO was more pronounced in patients with low nutritional status than in those with preserved nutritional reserves. This complicates the interpretation of the findings, as it remains unclear whether poor outcomes were primarily driven by high TMAO levels or by malnutrition, which is itself a known negative prognostic factor [67]. Additionally, B-type natriuretic peptide (BNP) levels, a recognized prognostic biomarker in heart failure, did not differ significantly between the high- and low-TMAO groups, limiting the strength of conclusions regarding the predictive value of TMAO alone [68].

### 7.3. TMAO, Heart Failure with Reduced and Mildly Reduced Ejection Fraction

In a study published in 2019 by Suzuki et al. [69], TMAO was found to be associated with a higher incidence of mortality and hospitalization in HF subjects with reduced (HEFrEF) or mildly reduced ejection fraction (HEFmrEF). TMAO and natriuretic peptide levels were evaluated based on the response to therapy according to the guidelines available at that time (beta-blockers, angiotensin-converting enzyme inhibitors/angiotensin receptor blockers, mineralocorticoid receptor antagonists, and diuretics). Although natriuretic peptides decreased after treatment, TMAO remained elevated, suggesting that there is no influence of the treatment on TMAO. This needs to be addressed in a future study, which would use a treatment regimen in accordance with the updated guidelines, including an angiotensin receptor and neprilysin inhibitor (ARNI) and a sodium-glucose co-transporter inhibitor (SGLT2-i) [69,70].

Although numerous studies have demonstrated that elevated TMAO levels are associated with poorer prognosis, including increased mortality and hospitalization rates [64,71,72], it remains unclear whether this relationship is causative. Furthermore, given TMAO’s renal excretion and the high prevalence of kidney dysfunction and systemic congestion in HF patients, it is uncertain whether elevated TMAO reflects direct harm or merely underlying comorbidities. Even if future research clarifies the mechanisms by which TMAO contributes to adverse outcomes, further investigation is needed to determine whether its reduction could serve as a preventive strategy against decompensation. Serial TMAO measurements alongside congestion markers, assessment across different HF phenotypes, and evaluation of TMAO fluctuations with SGLT2i or ARNI administration are necessary to address potential confounding factors.

## 8. Interventions

### 8.1. TMAO Measurements and Prognostic Value

The accurate assessment of circulating TMAO levels is essential for evaluating its role as a potential biomarker in CVD. However, several pre-analytical and analytical factors may significantly influence measured concentrations, thereby affecting both reproducibility and interpretation. Issues such as fasting versus postprandial sampling, recent fish intake, diurnal variation, antibiotic or probiotic use, and inter-laboratory differences in liquid chromatography–tandem mass spectrometry (LC-MS/MS) calibration represent critical challenges [48,49,73,74,75]. These methodological considerations are summarized below (Box 1).

Box 1Pre-analytical and analytical considerations for TMAO measurement.Pre-analytical issues:Fasting vs. post-prandial sampling can markedly affect TMAO levels.Recent fish intake transiently elevates circulating TMAO.Diurnal variation may introduce intra-individual fluctuations.Antibiotics or probiotics alter gut microbiota and TMAO production.Analytical issues:LC-MS/MS requires careful calibration for accuracy.Inter-laboratory variability limits comparability across studies.Within-person reliability data are needed before clinical adoption.

Since TMAO levels depend on kidney function, associations with cardiovascular outcomes may be attenuated after adjusting for eGFR, urea, and creatinine, making renal adjustment crucial in prognostic interpretations [76]. Furthermore, fish intake increases circulating TMAO, yet it is consistently linked with cardiovascular protection through omega-3 fatty acids and other nutrients. This paradox suggests that the relationship between TMAO and cardiovascular risk is context-dependent rather than uniform [73,77].

As summarized above, elevated plasma TMAO levels are independently associated with cardiovascular outcomes, including ASCVD, HF, and CKD, even after adjustment for conventional risk factors [13,47,53]. Several cohorts demonstrated incremental predictive value, with modest improvements in discrimination (ΔC-index) and reclassification (NRI) when TMAO was added to established risk models [78,79,80]. However, heterogeneity in populations, adjustment strategies, and assay methodologies complicates interpretation. Thus, while clinical validity is supported, consistency and generalizability remain areas for further investigation. To the best of our knowledge, there is currently no evidence that measuring TMAO alters patient management or improves outcomes. No human randomized controlled trials have tested a “test-and-target” approach, in which TMAO is measured, modifiable pathways are intervened upon, and clinical benefit is demonstrated. Without such data, clinical use remains limited to research, and TMAO should currently be regarded as an emerging biomarker with established prognostic validity but unproven clinical utility.

### 8.2. Dietary Strategies and Gut Microbiota Modulation

The absence of specific drugs that directly lower TMAO levels requires a focused effort on exploring and optimizing indirect approaches to reduce its concentration and mitigate its impact. There is a clear link between the gut microbiome, TMAO levels, and both metabolic and CVD. At the root of this chain lies the microbial composition of the gut, which is heavily influenced by diet [37]. First, dietary strategies should focus on modulating the intake of specific TMAO precursors—choline and carnitine—instead of broadly restricting all animal foods. For example, red meat is rich in both choline and carnitine and can increase TMAO production, whereas fish also contains these precursors but provides cardioprotective nutrients and should not be avoided indiscriminately [73]. Intake of choline and carnitine should be minimized but not eliminated to prevent adverse effects such as cognitive decline and fatty liver disease, while also considering the gut microbial composition of the individual [37,81]. Secondly, gut microbial composition can be favorably modulated through dietary fiber and polyphenols, found in vegetables, legumes, fruits, nuts, and seeds, which may modestly reduce TMAO production and support overall gut health. Probiotic interventions have been studied for their potential to shift microbiome composition and TMAO levels, but clinical endpoints such as cardiovascular outcomes remain unproven [82,83]. Therefore, these strategies should be framed as precision nutrition approaches aimed at influencing TMAO metabolism, rather than as established cardioprotective interventions.

### 8.3. Berberine: Reversing TMAO and AT-II Effects

Wang et al. [84] conducted a study on a murine model of hypertension induced by AT-II and observed a significant decrease in blood pressure after the administration of berberine, a nutritional supplement that is used in humans as well [85]. In this study, berberine administration reversed AT-II-induced endothelial-dependent vasodilation, improved gut microbial composition, and lowered TMAO levels. Further pilot randomized clinical trial studies in humans are needed to clarify whether the improvement in blood pressure applies, and if so, if berberine influences hypertension provoked by other mechanisms, not only by hyperexpression of AT-II.

### 8.4. TMA Lyase Inhibitors: Promising Pharmacological Tools

Animal studies show promising results regarding TMA-lyase inhibitors [86]. 3,3-Dimethyl-1-butanol (DMB), a choline analog derived from red wine, inhibits choline trimethylamine-lyase (CutC), a key enzyme in TMA production, thereby lowering plasma TMAO levels, reducing foam cell formation, and attenuating aortic plaque progression in mice, without altering circulating cholesterol [87]. Similar findings were reported for iodomethylcholine (IMC) and fluoromethylcholine (FMC). These compounds have minimal systemic absorption and exert their effects locally within the intestines, at the site of TMA production, and thereby more potently inhibit intestinal TMA production and reduce platelet activation without increasing bleeding risk [88].

Despite these encouraging preclinical data, human studies are currently lacking. Key translational considerations include characterization of pharmacokinetics and pharmacodynamics, demonstration of target engagement, evaluation of potential off-target effects (e.g., on platelet function), and assessment of antimicrobial resistance risk due to gut microbiome modulation. Therefore, while TMA-lyase inhibitors represent a promising therapeutic avenue, clinical efficacy and safety remain to be established in phase 1 and 2 studies.

### 8.5. Gene Silencing Therapy

Subjects with genetic mutations in the FMO3 gene exhibit abnormally high TMA levels due to reduced conversion to TMAO, clinically manifesting as trimethylaminuria (fish-odor syndrome) [89]. Bennet et al. [90] demonstrated that FMO3 is the most active enzyme in the FMO family for TMA oxidation in mice, with upregulation increasing TMAO levels and antisense oligonucleotide (ASO)–mediated FMO3 silencing reducing hepatic FMO3 mRNA by 90%, leading to a two-fold rise in TMA and a 47% decrease in plasma TMAO. These findings highlight the potential of FMO3 ASO as a strategy to lower TMAO. However, several translational unknowns remain: it is unclear whether similar reductions occur in humans, whether they translate into meaningful cardiovascular benefit, and whether TMA accumulation could cause adverse effects, including fish-odor syndrome or hepatic off-target consequences. At present, FMO3-targeted antisense oligonucleotide strategies remain exploratory, with long-term safety and clinical effects yet to be determined [91] (Figure 3).

## 9. Conclusions

The association between TMAO and cardiovascular disease has consistently provided valuable insights; however, evidence from human studies remains insufficient to establish a definitive causal relationship. While experimental models in mice support a potential causative role for TMAO, further clinical research is needed to validate these findings in humans and to clarify its relevance in cardiovascular pathology. Although therapeutic strategies targeting TMAO metabolism are currently under development and may become available in the coming years, additional efforts are required to determine whether TMAO is a reliable and clinically applicable biomarker for both gut dysbiosis and cardiovascular risk.

## Figures and Tables

**Figure 1 medicina-61-01767-f001:**
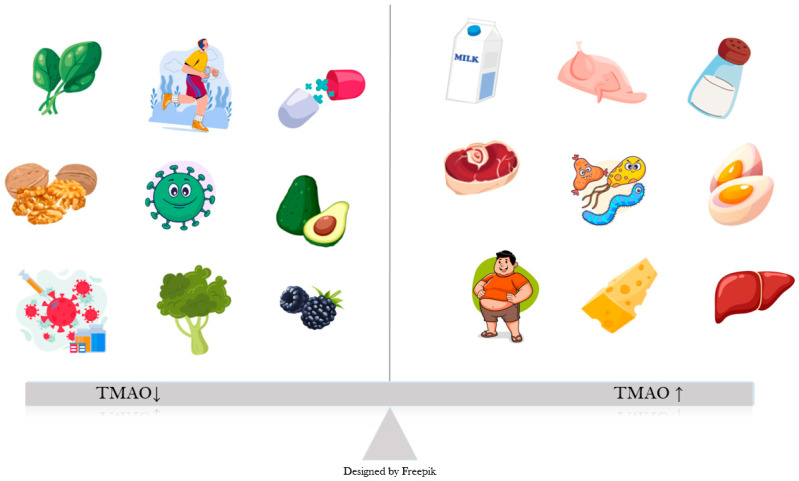
Factors influencing TMAO levels: Obesity, high salt intake, certain gut bacteria and animal sources of choline, such as meat, organs, dairy, eggs bacteria lead to an increase in plasma TMAO, as opposed to plant-based foods, exercise, probiotics and healthy gut bacteria.

**Figure 2 medicina-61-01767-f002:**
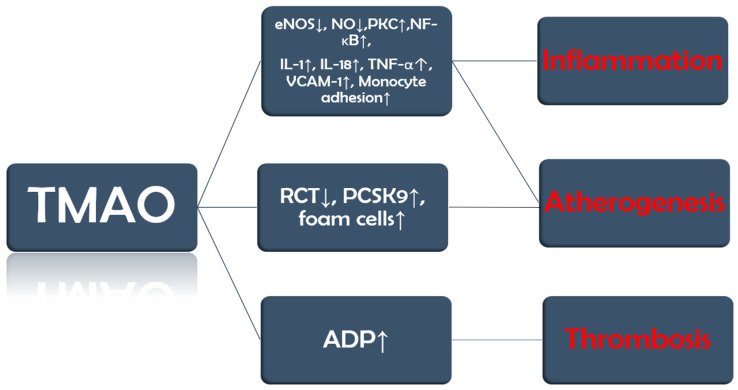
TMAO implications in CVD. The figure summarizes key concepts discussed in the text.

**Figure 3 medicina-61-01767-f003:**
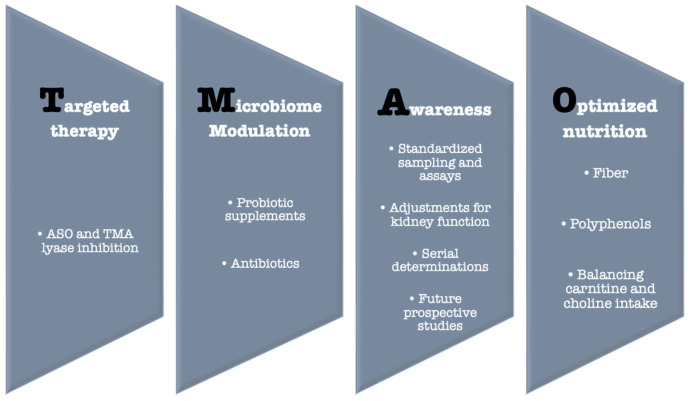
TMAO—Potential solutions.

## Data Availability

No new data were created or analyzed in this study. Data sharing is not applicable.

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
