# Peer review of "TMAO and Cardiovascular Disease: Exploring Its Potential as a Biomarker"

_medicina, 2025, doi:10.3390/medicina61101767_

Round 1

Reviewer 1 Report

Comments and Suggestions for Authors

The authors present a review entitled “TMAO and Cardiovascular Disease: Exploring Its Potential as a Biomarker”.

It should be noted that a review of on this topic is highly relevant. However, I have a few questions that need to be addressed:

  1. Page 3: “…as well as journals such as MDPI, the American Heart Journal, ScienceDirect, and Frontiers”. This includes publishers (MDPI, Frontiers), journal (AHJ), and database (SD).
  2. The study selection should be presented as a flowchart.
  3. The authors first state that the search for publications was limited to the last 10 years, then that for certain cases it was extended to 16 years. However, it should be clarified what “certain key information” required the extension of the time frame.
  4. Figure 1 requires clarification in the figure caption.

Author Response

Thank you for taking the time to revise our manuscript and for providing recommendations to help us improve this review.

Comment 1: Page 3: “…as well as journals such as MDPI, the American Heart Journal, ScienceDirect, and Frontiers”. This includes publishers (MDPI, Frontiers), journal (AHJ), and database (SD).

Response 1: Thank you for this observation! We have now categorized the sources accordingly.

Comment 2: The study selection should be presented as a flowchart.

Response 2: Thank you for your observation! Since our work is a narrative review, with the primary aim of providing a broad, integrative synthesis of the literature rather than a systematic appraisal, we decided not to include a flow diagram of study selection and removed the statement that “this review follows PRISMA guidelines.”

Comment 3: The authors first state that the search for publications was limited to the last 10 years, then that for certain cases it was extended to 16 years. However, it should be clarified what “certain key information” required the extension of the time frame.

Response 3: Thank you for your observation! We have clarified what “certain key information” referred to (see Section 2.3).

Comment 4: Figure 1 requires clarification in the figure caption.

Response 4: Thank you for your observation! We have added clarification in the figure caption.

Reviewer 2 Report

Comments and Suggestions for Authors

I appreciate the clinically framed question, “Can TMAO function as a biomarker and therapeutic target in CVD?” and the clear explanation of its origins, pathways, and disease connections. This makes the piece accessible to both clinicians and researchers. The manuscript effectively synthesizes recent animal and human data across hypertension, atherosclerosis, obesity/metabolic disease, and heart failure, connecting them to potential interventions like diet, probiotics, berberine, lyase inhibition, and FMO3 silencing. This comprehensive approach is valuable for readers entering the field. However, some aspects need clarification to ensure scientific rigor and transparency.

First, clarify the review type and methods. While you describe a narrative review performed “in accordance with PRISMA,” you don’t provide a protocol, flow diagram, or RoB assessment. Either reclassify it as a narrative review or upgrade it to a methods-complete systematic review with a protocol, PROSPERO registration, full search strings, dual screening, PRISMA flow diagram, and RoB tools per study design.

Ensure search reproducibility if you are going for a more formal review. Specify the exact database queries (Boolean strings, dates, filters) and justify the inclusion of Google Scholar and journal websites. Otherwise, readers cannot reproduce or evaluate the selection.

It would be valuable to assess study quality and grading. Include a structured evidence table (design, sample size, population, exposure/assay, outcomes, confounding factors, effect size, limitations) and grade the certainty of each clinical claim before drawing translational conclusions. This way, your claims can stand on their own merits.

As a researcher, emphasize the pivotal role of kidney function as a confounder. Highlight that TMAO is renally excreted, and associations with CVD diminish after adjusting for eGFR, urea, and creatinine. Make renal adjustment explicit in prognostic claims.

Fish raises circulating TMAO levels but is cardioprotective. Emphasize this paradox and caution against equating higher TMAO levels with uniformly higher risk outside cardiometabolic or renal contexts. This is crucial for biomarker positioning and dietary guidance.

Include a boxed note on pre-analytical and analytical issues, such as fasting vs. post-prandial sampling, recent fish intake, diurnal variation, antibiotics/probiotics, LC-MS/MS calibration, inter-lab variability, and the need for within-person reliability data before clinical adoption.

Extend the section on MR linking the microbiome to TMAO to summarize MR and large-scale genomic evidence investigating TMAO’s causality in CVD, including null or weak findings. Balance the narrative and clarify the distinction between correlation and causation. State where the evidence is insufficient.

For hypertension, emphasize that the 2020 meta-analysis pooled high-risk cohorts with limited dietary data, making generalizability to primary prevention uncertain. Soften mechanistic claims and highlight the need for prospective studies with diet/eGFR adjustment and standardized sampling.

When discussing atherosclerosis, separate plaque burden, plaque instability, and events and qualify each claim with design and adjustment details. The correlation with the SYNTAX score and OCT rupture is intriguing but not yet practice-changing. Explicitly state this.

Where you list effects like “decreases resting metabolic rate / increases appetite” anchor each to mechanistic or interventional human data. If the evidence is preclinical or indirect, label it as such or remove it to avoid overreach, otherwise you are just speculating!

About HF, try to strengthen the message that prognostic associations are consistent but mechanistic independence from renal function, GNRI and congestion is unresolved; articulate a practical research agenda (serial TMAO vs congestion markers; interaction with SGLT2i/ARNI; phenotypes HFpEF vs HFrEF), no?

Another thing, is to reorganize the “biomarker” argument under analytical validity then clinical validity than clinical utility. For analytical validity you need to show assay type, precision, thresholds, biological variation (add a table maybe), for clinical validity show independent, incremental prediction beyond standard risk (ASCVD, eGFR, hs-CRP, lipids), calibration, reclassification (ΔC-index, NRI), no? And for clinical utility try to think in a way that answers the question: does measuring TMAO impact management or outcomes? (Currently, there are no RCTs; propose test-and-target trials, to show gaps). Also clearly state that TMAO remains an emerging risk marker with limited utility evidence… it is what it is, no?

Reframe dietary guidance as precision nutrition, focusing on precursors and microbiota, rather than simply reducing animal foods. Distinguish choline/carnitine in red meat versus fish, warn against choline deficiency, and emphasize fiber and polyphenol strategies with realistic effect sizes on TMAO (acknowledging heterogeneity). Clarify probiotics endpoints (microbiome composition, TMAO levels, clinical outcomes) and avoid implying cardiovascular benefits without trial data. Berberine has pleiotropic effects and lacks human hypertension data in Ang-II phenotypes. Recommend pilot RCTs with pharmacovigilance before positioning it as an anti-TMAO strategy. Provide a translational roadmap for TMA-lyase inhibitors (human PK/PD, target engagement, platelet function off-targets, antimicrobial-resistance considerations). Avoid over-promising before phase 1/2 data. Show translational unknowns for FMO3 antisense (fish-odor syndrome risk, hepatic off-targets, long-term safety) and present it as hypothesis-generating only. Add renal handling, eGFR arrows, the fish paradox, and sampling/assay influencers (diet, antibiotics) to figures 1–3. Integrate mechanisms into a single pathway figure with intervention “levers.” Source every listed effect in table 1, separate human versus animal evidence, and clearly mark conjectural pathways. Remove items without direct support or relegate them to preclinical studies. Address inconsistencies, such as “6 metabolic diseases” appearing after “6.3” and “HEFrEF/HEFmREF” spellings. 

When stating that TMAO is “strongly associated” with CVD, consistently qualify it by population (CAD clinic vs. general population), adjustment sets (diet, kidney), and study design. Avoid implying causality. The cautious tone in the conclusion should be maintained throughout the body text.

Apologies for the long review. Incorporating these changes can improve the quality of your work. Looking forward to seeing the revised manuscript. 

Author Response

We sincerely thank you for the careful revision of our manuscript and for providing constructive recommendations that helped us improve the quality and clarity of this review.

Comment 1: First, clarify the review type and methods. While you describe a narrative review performed “in accordance with PRISMA,” you don’t provide a protocol, flow diagram, or RoB assessment. Either reclassify it as a narrative review or upgrade it to a methods-complete systematic review with a protocol, PROSPERO registration, full search strings, dual screening, PRISMA flow diagram, and RoB tools per study design. Ensure search reproducibility if you are going for a more formal review. Specify the exact database queries (Boolean strings, dates, filters) and justify the inclusion of Google Scholar and journal websites. Otherwise, readers cannot reproduce or evaluate the selection.

Response 1: We thank the reviewer for this important observation. Since our work is a narrative review, with the primary aim of providing a broad, integrative synthesis of the literature rather than a systematic appraisal, we removed the statement that “this review follows PRISMA guidelines.” In addition, we clarified and justified the inclusion of journal and publisher websites as sources (Section 2.1).

Comment 2: It would be valuable to assess study quality and grading. Include a structured evidence table (design, sample size, population, exposure/assay, outcomes, confounding factors, effect size, limitations) and grade the certainty of each clinical claim before drawing translational conclusions. This way, your claims can stand on their own merits.

Response 2: We appreciate your thoughtful remark regarding study quality and structured grading. While this type of structured appraisal is essential for systematic reviews, our manuscript was intentionally conceived as a narrative review, aiming to provide context, highlight conceptual connections, and critically synthesize the literature rather than formally grade each study. For this reason, we did not include an evidence table or certainty grading. Instead, we emphasized study design, methodological limitations, and interpretative cautions within the narrative, which we believe best fits the scope and objectives of this review.

Comment 3: As a researcher, emphasize the pivotal role of kidney function as a confounder. Highlight that TMAO is renally excreted, and associations with CVD diminish after adjusting for eGFR, urea, and creatinine. Make renal adjustment explicit in prognostic claims.

Response 3: Thank you for this valuable suggestion. We have added information regarding the relationship between kidney function and TMAO in a new subsection entitled “TMAO measurements and prognostic value” (Section 8.1).

Comment 4: Fish raises circulating TMAO levels but is cardioprotective. Emphasize this paradox and caution against equating higher TMAO levels with uniformly higher risk outside cardiometabolic or renal contexts. This is crucial for biomarker positioning and dietary guidance.

Response 4: We appreciate this observation. We added discussion of the “fish paradox” in Section 8.1.

Comment 5: Include a boxed note on pre-analytical and analytical issues, such as fasting vs. post-prandial sampling, recent fish intake, diurnal variation, antibiotics/probiotics, LC-MS/MS calibration, inter-lab variability, and the need for within-person reliability data before clinical adoption.

Response 5: Thank you for the suggestion. We added a text box summarizing pre-analytical and analytical considerations (section 8.1).

Comment 6: Extend the section on MR linking the microbiome to TMAO to summarize MR and large-scale genomic evidence investigating TMAO’s causality in CVD, including null or weak findings. Balance the narrative and clarify the distinction between correlation and causation. State where the evidence is insufficient.

Response 6: Thank you for this observation. We created a new subsection “3.3 Mendelian Randomization studies” where we summarize these results.

Comment 7: For hypertension, emphasize that the 2020 meta-analysis pooled high-risk cohorts with limited dietary data, making generalizability to primary prevention uncertain. Soften mechanistic claims and highlight the need for prospective studies with diet/eGFR adjustment and standardized sampling.

Response 7: Thank you for the suggestion. We revised the hypertension section accordingly to reflect these considerations.

Comment 8: When discussing atherosclerosis, separate plaque burden, plaque instability, and events and qualify each claim with design and adjustment details. The correlation with the SYNTAX score and OCT rupture is intriguing but not yet practice-changing. Explicitly state this.

Response 8: We thank the reviewer for this observation. We expanded the atherosclerosis section to distinguish these aspects and included study design details.

Comment 9: Where you list effects like “decreases resting metabolic rate / increases appetite” anchor each to mechanistic or interventional human data. If the evidence is preclinical or indirect, label it as such or remove it to avoid overreach, otherwise you are just speculating!

Response 9: Thank you for the suggestion. To avoid overreach, we decided to remove the table in Section 7.3.

Comment 10: About HF, try to strengthen the message that prognostic associations are consistent but mechanistic independence from renal function, GNRI and congestion is unresolved; articulate a practical research agenda (serial TMAO vs congestion markers; interaction with SGLT2i/ARNI; phenotypes HFpEF vs HFrEF), no?

Response 10: Thank you for this thoughtful point. We revised the end of Section 7.3 to incorporate these considerations.

Comment 11: Another thing, is to reorganize the “biomarker” argument under analytical validity then clinical validity than clinical utility. For analytical validity you need to show assay type, precision, thresholds, biological variation (add a table maybe), for clinical validity show independent, incremental prediction beyond standard risk (ASCVD, eGFR, hs-CRP, lipids), calibration, reclassification (ΔC-index, NRI), no? And for clinical utility try to think in a way that answers the question: does measuring TMAO impact management or outcomes? (Currently, there are no RCTs; propose test-and-target trials, to show gaps). Also clearly state that TMAO remains an emerging risk marker with limited utility evidence… it is what it is, no?

Response 11: Thank you for this observation. We reorganized the biomarker section accordingly. In Section 8.1 we added clinical validity and clinical utility (analytical validity is covered in the text box above).

Comment 12: Reframe dietary guidance as precision nutrition, focusing on precursors and microbiota, rather than simply reducing animal foods. Distinguish choline/carnitine in red meat versus fish, warn against choline deficiency, and emphasize fiber and polyphenol strategies with realistic effect sizes on TMAO (acknowledging heterogeneity). Clarify probiotics endpoints (microbiome composition, TMAO levels, clinical outcomes) and avoid implying cardiovascular benefits without trial data. Berberine has pleiotropic effects and lacks human hypertension data in Ang-II phenotypes. Recommend pilot RCTs with pharmacovigilance before positioning it as an anti-TMAO strategy. Provide a translational roadmap for TMA-lyase inhibitors (human PK/PD, target engagement, platelet function off-targets, antimicrobial-resistance considerations). Avoid over-promising before phase 1/2 data. Show translational unknowns for FMO3 antisense (fish-odor syndrome risk, hepatic off-targets, long-term safety) and present it as hypothesis-generating only.  

Response 12:  Thank you for these detailed and constructive suggestions. We revised section 8, expanding the discussion on diet, probiotics, berberine, TMA-lyase inhibitors, and FMO3 antisense as recommended. 

Comment 13: Add renal handling, eGFR arrows, the fish paradox, and sampling/assay influencers (diet, antibiotics) to figures 1–3. Integrate mechanisms into a single pathway figure with intervention “levers.”  

Response 13: Thank you for this helpful suggestion. Figure 3 was revised to incorporate adjustments for renal function, the need for standardized sampling/assays, and future prospective studies. While we attempted to integrate all mechanisms into a single master figure, this proved visually overcrowded, so we opted to distribute the information across multiple figures for clarity.

Comment 14:Source every listed effect in table 1, separate human versus animal evidence, and clearly mark conjectural pathways. Remove items without direct support or relegate them to preclinical studies.

Response 14: Thank you for the observation. We decided to remove Table 1, as it added limited value and the manuscript is already extensive.

Comment 15: Address inconsistencies, such as “6 metabolic diseases” appearing after “6.3” and “HEFrEF/HEFmREF” spellings.

Response 15: Thank you for pointing this out. We corrected the subsection numbering and the spelling of HFmrEF.

Comment 16: When stating that TMAO is “strongly associated” with CVD, consistently qualify it by population (CAD clinic vs. general population), adjustment sets (diet, kidney), and study design. Avoid implying causality. The cautious tone in the conclusion should be maintained throughout the body text.

Response 16: Thank you for this observation. We removed the term “strongly” from the relevant statements to avoid implying causality and to maintain a consistent cautious tone throughout the manuscript.

Round 2

Reviewer 2 Report

Comments and Suggestions for Authors

Overall, you have sufficiently addressed my concerns. While the work does not provide systematic evidence grading, the choice of a narrative format is justified and clearly stated. The revisions have improved clarity, balance and scientific caution. I consider the manuscript suitable for acceptance, pending only minor editorial checks.